# Plasma proteins associated with cardiovascular death in patients with chronic coronary heart disease: A retrospective study

**Lars Wallentin**[1,2]*, **Niclas Eriksson**[2], **Maciej Olszowka**[1,2], **Tanja B. Grammer**[3], **Emil Hagström**[1,2], **Claes Held**[1,2], **Marcus E. Kleber**[4], **Wolfgang Koenig**[5,6,7], **Winfried März**[4,8,9], **Ralph A. H. Stewart**[10], **Harvey D. White**[10], **Mikael Åberg**[11,12], **Agneta Siegbahn**[2,11,12]*

**1** Department of Medical Sciences, Cardiology, Uppsala University, Uppsala, Sweden, **2** Uppsala Clinical Research Center (UCR), Uppsala University, Uppsala, Sweden, **3** Mannheim Institute of Public Health, Social and Preventive Medicine, Mannheim Medical Faculty, University of Heidelberg, Heidelberg, Germany, **4** Medical Clinic V, Medical Faculty Mannheim, University of Heidelberg, Heidelberg, Germany, **5** Deutsches Herzzentrum München, Technische Universität München, Munich, Germany, **6** DZHK (German Centre for Cardiovascular Research), partner site Munich Heart Alliance, Munich, Germany, **7** Institute of Epidemiology and Medical Biometry, University of Ulm, Ulm, Germany, **8** Clinical Institute of Medical and Chemical Laboratory Diagnostics, Medical University of Graz, Graz, Austria, **9** SYNLAB Academy, SYNLAB Holding Deutschland GmbH, Mannheim and Augsburg, Germany, **10** Green Lane Cardiovascular Service, Auckland City Hospital and University of Auckland, Auckland, New Zealand, **11** Department of Medical Sciences, Clinical Chemistry, Uppsala University, Uppsala, Sweden, **12** Science for Life Laboratory, Uppsala University, Uppsala, Sweden

* lars.wallentin@ucr.uu.se (LW); Agneta.Siegbahn@medsci.uu.se (AS)

**Data Availability Statement:** Anonymized individual participant data and study documents can be requested for further research from www.clinicalstudydatarequest.com.

## Abstract

### Background

Circulating biomarkers are associated with the development of coronary heart disease (CHD) and its complications by reflecting pathophysiological pathways and/or organ dysfunction. We explored the associations between 157 cardiovascular (CV) and inflammatory biomarkers and CV death using proximity extension assays (PEA) in patients with chronic CHD.

### Methods and findings

The derivation cohort consisted of 605 cases with CV death and 2,788 randomly selected non-cases during 3–5 years follow-up included in the STabilization of Atherosclerotic plaque By Initiation of darapLadIb TherapY (STABILITY) trial between 2008 and 2010. The replication cohort consisted of 245 cases and 1,042 non-cases during 12 years follow-up included in the Ludwigshafen Risk and Cardiovascular Health (LURIC) study between 1997 and 2000. Biomarker levels were measured with conventional immunoassays and/or with the OLINK PEA panels CVD I and Inflammation. Associations with CV death were evaluated by Random Survival Forest (RF) and Cox regression analyses.

Both cohorts had the same median age (65 years) and 20% smokers, while there were slight differences in male sex (82% and 76%), hypertension (70% and 78%), and diabetes (39% and 30%) in the respective STABILITY and LURIC cohorts. The analyses identified 18

**Funding:** The present study was supported by the Swedish Foundation for Strategic Research (project RB13-0197) to LW and AS, and by Science for Life Laboratory, Uppsala, Sweden (https://www.scilifelab.se/). Funding from own institution, Uppsala Clinical Research Center, Uppsala, Sweden (Sweden Organisation number 232100-0024) was given to LW and AS. OLINK Proteomics and Roche Diagnostics provided their respective assays at a reduced cost. GlaxoSmithKline sponsored the main STABILITY trial and the biobanking of the samples but provided no specific support for this sub-study. None of the sponsoring companies had any input on the study design, analyses, interpretation or manuscript preparation. The sponsor was given the opportunity to review and comment on the manuscript. LURIC received funding from the European Union's Horizon 2020 research and innovation programme under the ERA-Net Cofund action N˚ 727565 (OCTOPUS project) and the German Ministry of Education and Research (grant number 01EA1801A), from the 7th Framework Program (integrated projects AtheroRemo, Grant Agreement number 201668 and RiskyCAD, Project Number 305739) of the European Union, and the Competence Cluster of Nutrition and Cardiovascular Health (nutriCARD) which is funded by the German Federal Ministry of Education and Research (grant number 01EA1411A). None of the sponsors had any input on the study design, analyses, interpretation, the decision to publish, or preparation of the manuscript.

**Competing interests:** I have read the journal's policy and the authors of this manuscript have the following competing interests: LW reports institutional research grants from AstraZeneca, Boehringer Ingelheim, Bristol-Myers Squibb/Pfizer, GlaxoSmithKline, Merck & Co, Roche Diagnostics; consulting fees from Abbott; holds two patents involving GDF-15, both licensed to Roche Diagnostics. NE and MO report institutional research grants from GlaxoSmithKline. EH reports institutional research grants from AstraZeneca, Amgen, GlaxoSmithKline, Sanofi; consulting fees from Amgen, NovoNordisk, Sanofi; speaker fees from AstraZeneca, Amgen, Boehringer Ingelheim, NovoNordisk, Sanofi; principal investigator fees paid to institution from CSL Behring, Dalcor, Regeneron, Sanofi, The Medicines Company. WK reports research grants from Roche Diagnostics, Beckmann, Singulex, Abbott; other research support from European Research Agency (ERA-CVD); honoraria from Novartis, Pfizer, Sanofi, AstraZeneca, Amgen; expert witness fees from Novartis; consultant/advisory board fees from Novartis, Pfizer, DalCor, Sanofi, Kowa, Amgen. CH

biomarkers with confirmed independent association with CV death by Boruta analyses and statistical significance (all $p < 0.0001$) by Cox regression when adjusted for clinical characteristics in both cohorts. Most prognostic information was carried by N-terminal prohormone of brain natriuretic peptide (NTproBNP), hazard ratio (HR for 1 standard deviation [SD] increase of the log scale of the distribution of the biomarker in the replication cohort) 2.079 (95% confidence interval [CI] 1.799–2.402), and high-sensitivity troponin T (cTnT-hs) HR 1.715 (95% CI 1.491–1.973). The other proteins with independent associations were growth differentiation factor 15 (GDF-15) HR 1.728 (95% CI 1.527–1.955), transmembrane immunoglobulin and mucin domain protein (TIM-1) HR 1.555 (95% CI 1.362–1.775), renin HR 1.501 (95% CI 1.305–1.727), osteoprotegerin (OPG) HR 1.488 (95% CI 1.297–1.708), soluble suppression of tumorigenesis 2 protein (sST2) HR 1.478 (95% CI 1.307–1.672), cystatin-C (Cys-C) HR 1.370 (95% CI 1.243–1.510), tumor necrosis factor-related apoptosis-inducing ligand receptor 2 (TRAIL-R2) HR 1.205 (95% CI 1.131–1.285), carbohydrate antigen 125 (CA-125) HR 1.347 (95% CI 1.226–1.479), brain natriuretic peptide (BNP) HR 1.399 (95% CI 1.255–1.561), interleukin 6 (IL-6) HR 1.478 (95% CI 1.316–1.659), hepatocyte growth factor (HGF) HR 1.259 (95% CI 1.134–1.396), spondin-1 HR 1.295 (95% CI 1.156–1.450), fibroblast growth factor 23 (FGF-23) HR 1.349 (95% CI 1.237–1.472), chitinase-3 like protein 1 (CHI3L1) HR 1.284 (95% CI 1.129–1.461), tumor necrosis factor receptor 1 (TNF-R1) HR 1.486 (95% CI 1.307–1.689), and adrenomedullin (AM) HR 1.750 (95% CI 1.490–2.056).

The study is limited by the differences in design, size, and length of follow-up of the 2 studies and the lack of results from coronary angiograms and follow-up of nonfatal events.

## Conclusions

Profiles of levels of multiple plasma proteins might be useful for the identification of different pathophysiological pathways associated with an increased risk of CV death in patients with chronic CHD.

## Trial registration

ClinicalTrials.gov NCT00799903.

## Author summary

### Why was this study done?

- There are many reports on associations between biomarkers and outcomes in patients with chronic coronary artery disease (CAD).
- New analytical technologies allow concurrent measurement of hundreds of protein biomarkers in small volumes of plasma.
- The value of multiplex protein analyses has rarely been evaluated in cohorts with adequate numbers of patients and outcome events.

reports Honoraria from Pfizer; consultant/advisory board fees from AstraZeneca, Bayer, Boehringer Ingelheim. MEK reports consulting/speaker fees from Bayer. WM reports grants and personal fees from Siemens Diagnostics, Abbott Diagnostics, AMGEN, Astrazeneca, Sanofi, BASF, Numares AG, employment from SYNLAB Holding Deutschland GmbH. RAHS reports grants and non-financial support from GlaxoSmithKline. HDW reports research grants from Sanofi-Aventis, Dalcor Pharma, NIH, Eisai, Omthera Pharmaceuticals; other support from SAHMRI, ESC, AHA, Sanofi/Regeneron. AS reports institutional research grants from AstraZeneca, Boehringer Ingelheim, Bristol-Myers Squibb/Pfizer, Roche Diagnostics, GlaxoSmithKline; consulting fees from Olink Proteomics. TBG and MÅ have nothing to declare.

**Abbreviations:** AM, adrenomedullin; BMI, body mass index; BNP, brain natriuretic peptide; CA-125, carbohydrate antigen 125; CABG, coronary artery bypass grafting; CAD, coronary artery disease; CHD, coronary heart disease; CHI3L1, chitinase-3 like protein 1; CI, confidence interval; CRP-hs, high-sensitivity C-reactive protein; cTnT-hs, high-sensitivity troponin T; CV, cardiovascular; Cys-C, cystatin-C; FGF-23, fibroblast growth factor 23; GDF-15, growth differentiation factor 15; HGF, hepatocyte growth factor; HR, hazard ratio; IL-6, interleukin 6; LDL, low-density lipoprotein; LLOD, lower limit of detection; Lp-PLA$_2$, lipoprotein associated phospholipase A2; LURIC, Ludwigshafen Risk and Cardiovascular Health; maxstat, maximally selected rank statistics; MI, myocardial infarction; NPX, normalized protein expression; NTproBNP, N-terminal prohormone of brain natriuretic peptide; OPG, osteoprotegerin; PCI, percutaneous coronary intervention; PEA, proximity extension assay; RAS, renin–angiotensin system; RF, Random Survival Forest; SD, standard deviation; sST2, soluble suppression of tumorigenesis 2 protein; STABILITY, STabilization of Atherosclerotic plaque By Initiation of darapladIb TherapY; TIM-1, transmembrane immunoglobulin and mucin domain protein; TIMP-1, tissue inhibitor of metalloproteinase-1; TNF-R1, tumor necrosis factor receptor 1; TRAIL-R2, tumor necrosis factor-related apoptosis-inducing ligand receptor 2.

## What did the researchers do and find?

- We investigated the associations between the levels of multiple proteins and the occurrence of cardiovascular (CV) death during 3–12 years follow-up of 2 cohorts of 3,393 and 1,287 patients with chronic CAD.

- The biomarkers were measured with the OLINK Proximity Extension Assay (PEA) panels CVD I and Inflammation and/or conventional immunoassays.

- The analyses identified 18 biomarkers with confirmed independent associations with CV death.

## What do these findings mean?

- Measurements of levels of plasma protein profiles can be useful for the identification of pathophysiological pathways associated with an increased risk of CV death in patients with chronic coronary heart disease (CHD).

- Measurements of these profiles might be useful for the identification of new treatment targets and to balance different treatments and treatment responses in patients with chronic CHD.

## Introduction

Despite revascularization and optimal secondary preventive treatment, chronic coronary heart disease (CHD) is still associated with a substantial cardiovascular (CV) morbidity and mortality [1–3]. Many biomarkers have been shown to be associated with the development and clinical outcomes of CHD. However, only a few biomarkers are recommended in treatment guidelines and routinely used in clinical care [4–8]. Besides indicators of underlying metabolic disease (blood glucose, hemoglobin A1c, and low-density lipoprotein [LDL] cholesterol), the currently generally available prognostic protein biomarkers are those indicating myocardial dysfunction (N-terminal prohormone of brain natriuretic peptide, NTproBNP) [6,7], cardiac necrosis (cardiospecific high-sensitivity troponin T, cTnT-hs), inflammatory activity (high-sensitivity C-reactive protein, CRP-hs) [7], and renal dysfunction (cystatin-C, Cys-C) [4,8]. The rationale for the use of these biomarkers is their associations with underlying disease mechanisms, their incremental prognostic information on CV outcomes in addition to clinical information, and their interaction with treatment effects. There are many reports on associations between a multitude of other biomarkers and the development of CHD. However, their value has rarely been simultaneously evaluated in cohorts with adequate numbers of patients and events and including adjustment both for clinical information and for the established biomarkers.

Newly available analytical technologies allow concurrent measurement of hundreds and even thousands of protein biomarkers in small volumes of plasma [9]. One of these techniques is the proximity extension assay (PEA) technology, allowing simultaneous measurements of 92 proteins in a 1.0-μl plasma sample by PCR amplification of DNA strands from DNA-labeled antibody pairs [10,11]. In the STabilization of Atherosclerotic plaque By Initiation of darapLadIb TherapY (STABILITY) trial (ClinicalTrials.gov ID NCT00799903) randomizing 15,828

patients with chronic CHD to darapladib or placebo, without any significant effect on outcomes during 3 to 5 years follow-up, we obtained plasma aliquots from all patients at baseline [12,13]. Our previous reports from this program have verified the independent prognostic importance of NTproBNP, cTnT-hs, CRP-hs, interleukin 6 (IL-6), growth differentiation factor 15 (GDF-15), and lipoprotein associated phospholipase A2 (Lp-PLA$_2$) [14–17]. The primary aim of the present multimarker substudy was to investigate if additional biomarkers were associated with CV death in patients with chronic CHD. The findings were replicated in an observational cohort with chronic CHD followed for 12 years in the Ludwigshafen Risk and Cardiovascular Health (LURIC) study [16].

## Methods

### Patients and study cohorts

The initial cohort of the present study was a substudy of the previously published international STABILITY trial, which compared darapladib, a selective inhibitor of Lp-PLA$_2$, with placebo concerning the occurrence of CV events in 15,828 patients with chronic CHD out of whom 14,124 provided blood samples for the biomarker substudy [12,13]. The patients were recruited in 663 centers in 39 countries from December 2008 to April 2010. The patients were followed from 3 to 5 years with a median of 3.7 years. Patients were eligible if they had chronic CHD documented by prior myocardial infarction (MI) (>1 month), prior percutaneous coronary intervention (PCI) (>1 month), or prior coronary artery bypass grafting (CABG) (>3 months) or multivessel coronary artery disease (CAD) at a coronary angiogram. Outcomes were ascertained by regular follow-up visits until the end of the trial, and all study endpoints were centrally adjudicated. There were no significant effects of the randomized treatment on the outcomes. The present multimarker substudy was based on an unstratified case–cohort design consisting of a random sample from the full cohort, which was enriched with all patients suffering CV outcomes in the total biomarker cohort leading to a comparison between 2,788 patients without CV death and 605 patients with CV death.

The LURIC prospective observational study was used for replication. The LURIC study included 3,316 patients scheduled for coronary angiography between July 1997 and January 2000. Patients presenting with unstable angina, non-ST elevation MI, ST-elevation MI, and severe diseases other than chronic CHD were excluded in the current study. The patients were followed for 12 years concerning vital status, and cause of death was established by death certificates. In these analyses, 245 patients with CV death during the 12 years follow-up constituted the cases and 1,042 patients without this event the non-cases.

### Ethics statement

Both studies were approved by the relevant institutional review boards and performed in accordance with the Declaration of Helsinki, and all patients provided written informed consent [16].

### Biochemical analyses

Venous blood samples were obtained at inclusion in the morning after 9 hours of fasting. EDTA plasma aliquots were stored at −80˚C until biochemical analyses. Established biochemical assays for NTproBNP, cTnT-hs, CRP-hs, IL-6, GDF-15, and Lp-PLA$_2$ were centrally performed as previously published [12–17]. The proteomic analyses were performed at the Clinical Biomarkers Facility, Science for Life Laboratory, Uppsala University, Uppsala, Sweden, without information on any other data. We used the OLINK Proteomics PEA technology,

which is based on pairs of antibodies equipped with DNA single-strand oligonucleotide reporter molecules. Each OLINK PEA panel contains 96 oligonucleotide-labeled antibody probe pairs that bind to their respective target if present in the sample [9,10]. Target binding by both antibodies in a pair generates double-stranded DNA amplicons, which are quantified using a Fluidigm BioMark™ HD real-time PCR platform. The analyses were run using the recommended internal control, and inter-plate variability was adjusted by intensity normalization. The resulting relative values, normalized protein expression (NPX) data, were log2 transformed. In the logarithmic phase of the curve, 1 increase of the NPX value corresponds to a doubling of the protein content, and a high NPX value corresponds to a high protein concentration. The signal specificity is exceptionally high, as binding by the 2 protein specific antibodies in close proximity is required to produce a signal. PEA assays have shown high reproducibility and repeatability with mean intra-assay and inter-assay coefficients of variation around 8% and 12%, respectively; average inter-site variation has been reported at 15% [9]. In the STABILITY cohort, we used the OLINK Proteomics Multiplex CVD I$^{96\times96}$ panel and the OLINK Proteomics Multiplex Inflammation$^{96\times96}$ panel, whereby the concentrations of a total of 157 proteins related to CV disease and inflammation were simultaneously measured. In the LURIC cohort, we used only the OLINK Proteomics Multiplex CVD I$^{96\times96}$ panel as only 4 of the 28 biomarkers with confirmed association to CV death in the STABILITY cohort were in the Inflammation panel and none among the 13 most important. The protein markers included in the CVD I and Inflammation panels are given in S1 Table.

## Statistics

The PEA CVD I and Inflammation panels together measured the concentrations of 184 proteins, of which 27 were included in both panels. For 32 proteins, more than 10% of patient samples had levels below the lower limit of detection (LLOD). We chose to include as many proteins as possible by excluding only the 3 proteins for which more than 99.5% of all patient samples had values below LLOD, leaving 154 unique proteins for the statistical analyses (S2A Table). Values below LLOD were imputed with LLOD/2.

Baseline characteristics, established biomarkers, and PEA biomarkers were presented and compared by conventional statistics: chi-squared and Wilcoxon tests for discrete and continuous variables, respectively. Correlations between conventionally measured biomarkers and PEA biomarkers were presented as Spearman correlation coefficients using NPX data for PEA biomarkers and logarithmic transformation of biomarker concentrations measured with other assays. Because of the large number of observations, most correlation coefficients were expected to be statistically significant, and only correlation coefficients above 0.29 explaining at least 9% of the variability were considered relevant. Correlations between PEA biomarkers present on both CVD I and Inflammation panels were calculated using Pearson correlation coefficients.

Random Survival Forest (RF) analyses were performed to provide an unbiased grading of the prognostic importance of all variables including all clinical characteristics, PEA levels, and levels of other biomarkers. As the variable importance calculated by the RF could theoretically give identical results in a sample of 1,000 to 100,000 individuals, a Boruta analysis was used to confirm which of the variables in the RF analysis had a larger than random association with outcomes [18]. In short, the Boruta analysis performs multiple runs of RF comparing all variables to random variables, which are shuffled copies of the original variables. Variables performing better than the maximum random variable importance are classified as confirmed, variables performing worse are rejected, and variables that cannot be confirmed or rejected are classified as tentative. Biomarkers with confirmed associations at RF-Boruta analyses in both

studies were considered to have an externally validated association with CV death. In both RF and Boruta analyses, the following settings were used: variable importance mode = permutation, mtry (randomly selected number of variables to possibly split in each node) = square root of total number of variables, minimal node size = 3, and splitrule = maxstat (maximally selected rank statistics). In the RF, we used 10,000 trees, and in the Boruta analysis, the number of trees was lowered to 1,000 due to performance issues.

The linear associations between the biomarkers and CV death were also investigated by Cox regression analyses including both clinical characteristics, established standard immunoassays, and PEA biomarkers and were presented as Forest plots. According to the study design, the Cox regression analyses included the sampling weights for each outcome and were estimated using a robust sandwich estimator. The Cox regression analyses were performed unadjusted using a predefined model [12–17] adjusting for baseline characteristics (age, sex, body mass index (BMI), current smoking, hypertension, diabetes mellitus, previous MI, previous coronary revascularization, previous stroke, previous peripheral artery disease, and randomized treatment) and adjusting also for renal function (Cys-C) and the established markers of CV risk (NTproBNP and cTnT-hs). The incremental discriminative value of each biomarker was illustrated by the C-index. If conventional as well as PEA measurements were available, only the conventional result was included in the RF and Cox analyses, considering its quantitative measurement and larger dynamic range. In the RF, Boruta, and Cox regression analyses, missing values (other than protein values below LLOD) were imputed once using the mice package for R [19,20]. The statistical analyses were predefined in a statistical analysis plan for the STABILITY PEA study in March 2017, and thereafter for replication also applied for the LURIC PEA study.

## Results

### PEA measurements

The PEA measurements had acceptable reproducibility with Pearson correlation coefficients 0.80 to 0.97 for 23 and a standard deviation of the difference in levels of 0.18 to 0.31 for 17 of the 27 proteins that were included in both the CVD I and Inflammation panels (S4A Table). The PEA measurements also had adequate accuracy with close correlations with conventional immunoassays with Spearman correlation coefficients of 0.87 for NTproBNP, 0.85 for GDF-15, and 0.88 for IL-6 and similar associations to CV death with both methods (S4B Table). The Spearman correlations between the immunoassays of the cardiorenal and inflammatory biomarkers Cys-C, NTproBNP, cTnT-hs, GDF-15, and IL-6 and the PEA biomarkers in the STABILITY cohort are shown in S5 Table for those PEA biomarkers with any correlation coefficient >0.29. There were significant and relevant correlations between many biomarkers, prompting multivariable adjustments when evaluating the importance of associations between biomarker levels and clinical outcomes.

### STABILITY population

Baseline characteristics in the random sample and in the total STABILITY trial were almost identical (S3 Table). The baseline characteristics and the levels of established CV biomarkers in the group with CV death and in the random sample without any such event are shown in Table 1.

The crude bivariate comparison of the 154 biomarkers between the 605 cases and 2,788 non-cases showed significant differences with $p < 0.0001$ for 87 biomarkers (S6A Table). An unbiased selection of variables (biomarkers as well as clinical variables) with linear or nonlinear associations with CV death occurrence was performed by RF analyses. Prognostic

**Table 1. Baseline characteristics and levels of established biomarkers in patients with and without CV death in the STABILITY and LURIC.**

| Baseline characteristics | STABILITY | | | LURIC | | |
|---|---|---|---|---|---|---|
| | No event (N = 2,788) | CV death (N = 605) | p-value | No event (N = 1,042) | CV death (N = 245) | p-value |
| Age (years) | 59/65/71 | 61/69/75 | <0.001 | 57/63/70 | 63/69/74 | <0.001 |
| Sex: Male | 82% (2,283) | 83% (502) | 0.527 | 75% (779) | 80% (196) | 0.085 |
| Smoker | 20% (549) | 22% (130) | 0.299 | 16% (169) | 16% (40) | 0.021[a] |
| Hypertension | 70% (1,950) | 78% (470) | <0.001 | 76% (794) | 82% (200) | 0.068 |
| Diabetes | 38% (1,054) | 53% (318) | <0.001 | 26% (225) | 43% (106) | <0.001 |
| Prior MI | 59% (1,644) | 67% (405) | <0.001 | 43% (444) | 55% (134) | <0.001 |
| Prior PCI or CABG | 76% (2,114) | 61% (369) | <0.001 | 37% (382) | 40% (97) | 0.403 |
| Prior stroke or TIA | 9% (241) | 11% (67) | 0.059 | 7% (70) | 19% (47) | <0.001 |
| Prior PAD | 8% (215) | 17% (105) | <0.001 | 10% (102) | 19% (47) | <0.001 |
| Randomized to darapladib | 52% (1,437) | 48% (290) | 0.108 | | | |
| **Medications at baseline** | | | | | | |
| Aspirin | 92% (2,572) | 90% (542) | 0.031 | 74% (766) | 72% (176) | 0.594 |
| P2Y12 inhibitors | 34% (936) | 30% (183) | 0.115 | | | |
| Beta-blockers | 79% (2,203) | 81% (488) | 0.366 | 65% (675) | 52% (127) | <0.001 |
| Statin treatment | 97% (2,709) | 98% (594) | 0.159 | 52% (546) | 48% (118) | 0.233 |
| ACE or angiotensin receptor inhibitors | 76% (2,124) | 80% (483) | 0.054 | 53% (554) | 70% (172) | <0.001 |
| **Established biomarkers** | | | | | | |
| GFR by CKD-EPI (ml/min) | 62/74/86 | 52/66/81 | <0.001 | 76/88/97 | 61/79/93 | <0.001 |
| Cys-C (mg/L) | 0.86/1.00/1.16 | 0.98/1.18/1.44 | <0.001 | 0.81/0.90/1.04 | 0.89/1.03/1.27 | <0.001 |
| NTproBNP (ng/L) | 85/174/353 | 250/573/1,269 | <0.001 | 94/230/616 | 292/762/1927 | <0.001 |
| cTnT-hs (ng/L) | 6.3/9.3/14.1 | 10.8/16.4/26.0 | <0.001 | 5.00/8.41/14.85 | 9.66/17.00/30.98 | <0.001 |
| CRP-hs (mg/L) | 0.6/1.3/3.0 | 0.9/1.8/4.7 | <0.001 | 1.120/2.410/5.805 | 1.590/4.420/10.200 | <0.001 |
| IL-6 (ng/L) | 1.4/2.1/3.2 | 1.9/3.0/4.9 | <0.001 | | | |
| GDF-15 (ng/L) | 914/1,243/1,766 | 1,231/1,755/2,738 | <0.001 | | | |
| Lp-PLA2 (μmol/min/L) | 144.3/171.5/201.7 | 150.0/183.6/219.2 | <0.001 | | | |
| LDL cholesterol (mmol/L) | 1.64/2.10/2.61 | 1.65/2.20/2.91 | 0.015 | 2.43/2.98/3.60 | 2.51/2.95/3.57 | 0.909 |
| HDL cholesterol (mmol/L) | 1.00/1.18/1.40 | 0.97/1.16/1.37 | 0.149 | 0.83/0.98/1.14 | 0.80/0.96/1.19 | 0.221 |
| Triglycerides (mmol/L) | 1.09/1.50/2.08 | 1.04/1.48/2.08 | 0.359 | 1.27/1.68/2.29 | 1.16/1.58/2.23 | 0.095 |
| Hemoglobin (g/L) | 134/144/152 | 130/141/152 | 0.001 | 131/141/150 | 126/137/148 | 0.003 |
| WBC count ($10^9$/L) | 5.4/6.5/7.7 | 6.0/7.2/8.3 | <0.001 | 5.59/6.60/7.90 | 5.80/6.88/8.40 | 0.022 |

Categorical variables are reported as % (n), whereas continuous variables are reported by the percentiles 25th/50th/75th.

CKD-EPI was calculated based on creatinine, age, and gender.

a Comparison also includes the category prior smoker, which were 50% (517) for non-cases and 58% (143) for cases in the LURIC data.

ACE, angiotensin converting enzyme; CABG, coronary artery bypass grafting; CKD-EPI, chronic kidney disease epidemiology collaboration; CRP-hs, high-sensitivity C-reactive protein; cTnT-hs, high-sensitivity troponin T; CV, cardiovascular; Cys-C, cystatin-C; GDF-15, growth differentiation factor 15; GFR, glomerular filtration rate; HDL, high-density lipoprotein; IL-6, interleukin 6; LDL, low-density lipoprotein; Lp-PLA2, lipoprotein associated phospholipase A2; LURIC, Ludwigshafen Risk and Cardiovascular Health; MI, myocardial infarction; NTproBNP, N-terminal prohormone of brain natriuretic peptide; PAD, peripheral artery disease; PCI, percutaneous coronary intervention; STABILITY, STabilization of Atherosclerotic plaque By Initiation of darapLadIb TherapY; TIA, transient ischemic attack; WBC, white blood cell.

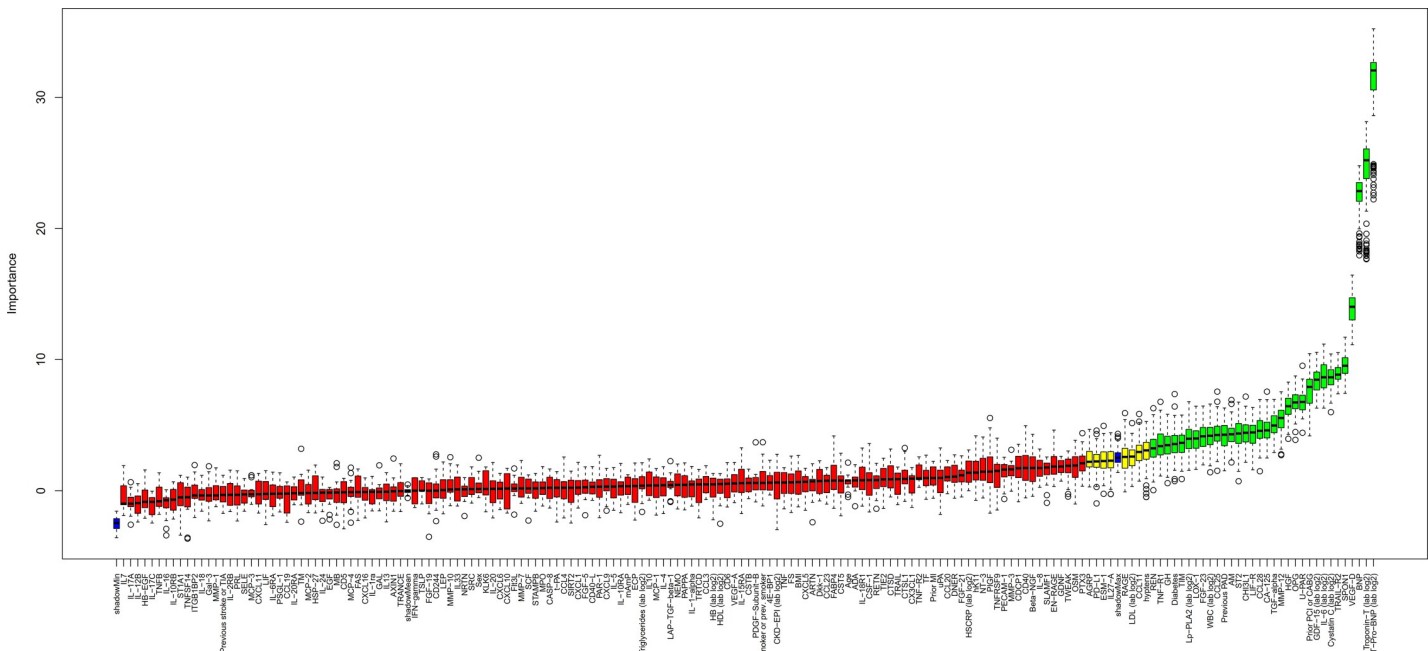

**Fig 1. Variable importance in the STABILITY cohort.** Boruta analysis in the STABILITY cohort of the significance of variable importance for CV death in the RF analysis, including clinical variables as well as established and PEA biomarkers. Values are NPX values for the PEA biomarkers and log2 for the ng/L levels of NTproBNP, cTnT-hs, IL-6, GDF-15, Cys-C, CRP-hs, Lp-PLA₂, and WBC measured by conventional quantitative assays. Color coding according to the Boruta analysis result: green = confirmed, yellow = tentative, and red = rejected. CRP-hs, high-sensitivity C-reactive protein; cTnT-hs, high-sensitivity troponin T; CV, cardiovascular; Cys-C, cystatin-C; GDF-15, growth differentiation factor 15; IL-6, interleukin 6; Lp-PLA₂, lipoprotein associated phospholipase A2; NPX, normalized protein expression; NTproBNP, N-terminal prohormone of brain natriuretic peptide; PEA, proximity extension assay; RF, Random Survival Forest; STABILITY, STabilization of Atherosclerotic plaque By Initiation of darapLadIb TherapY; WBC, white blood cell.

importance is presented in S1A Fig. According to the corresponding Boruta analysis, 28 biomarkers had confirmed importance for the selection of patients with CV death (Fig 1, Table 2). Linear associations of biomarker levels with CV death were investigated by unadjusted (S2A Fig) and adjusted Cox regression analyses (Fig 2, Table 2). According to these analyses, NTproBNP and cTnT-hs carried most of the prognostic information, and addition of any of the other biomarkers increased C-index by less than 0.01 (Table 2).

## Replication population

The baseline characteristics of the LURIC population is shown in Table 1. In this cohort, the RF and Boruta analyses identified 21 biomarkers of confirmed importance for CV death, of which 18 also had been identified in the STABILITY cohort (Figs 3 and 4, S1B and S2B Figs, S6B Table). Also, in the LURIC cohort, the 2 strongest prognostic biomarkers were NTproBNP and cTnT-hs. At the next level of importance, another 16 biomarkers of prognostic importance were verified, in descending order: GDF-15, adrenomedullin (AM), osteoprotegerin (OPG), transmembrane immunoglobulin and mucin domain protein (TIM-1), renin (REN), Cys-C, tumor necrosis factor (TNF)-related apoptosis-inducing ligand receptor 2 (TRAIL-R2), soluble suppression of tumorigenesis 2 protein (sST2), BNP, hepatocyte growth factor (HGF), IL-6, carbohydrate antigen 125 (CA-125), spondin 1 (SPON-1), tumor necrosis factor receptor 1 (TNF-R1), fibroblast growth factor 23 (FGF-23), and chitinase-3 like protein 1 (CHI3L1). When investigating these biomarkers by Cox regression analyses, adjusting for renal function (Cys-C) and the strongest prognostic biomarkers NTproBNP and cTnT-hs, only 1 biomarker (REN) increased the C-index by more than 0.01 (Table 3, Fig 4).

**Table 2. Associations between CV death and biomarkers RF analysis in the STABILITY cohort including biomarkers with confirmed importance according to the Boruta procedure and also showing results from the Cox regression analyses.**

| | Order RF | By SD | Adjusted by clinical variables[a] | | | | Also for Cys-C+NTproBNP+cTnT-hs[b] | | | |
|---|---|---|---|---|---|---|---|---|---|---|
| | | | HR | CI | C-index | p-value | HR | CI | C-index | p-value |
| NTproBNP | 1 | 1.74 | 2.846 | (2.524–3.209) | 0.802 | <0.001 | 2.348 | (2.051–2.689) | 0.808 | <0.001 |
| Troponin-T | 2 | 0.96 | 2.048 | (1.865–2.248) | 0.766 | <0.001 | 1.483 | (1.323–1.663) | 0.808 | <0.001 |
| BNP | 3 | 1.84 | 2.292 | (2.058–2.554) | 0.765 | <0.001 | 1.089 | (0.910–1.302) | 0.808 | 0.352 |
| VEGF-D | 4 | 0.53 | 1.933 | (1.673–2.234) | 0.735 | <0.001 | 1.261 | (1.089–1.459) | 0.810 | 0.002 |
| TRAIL-R2 | 5 | 0.49 | 1.255 | (1.149–1.370) | 0.722 | <0.001 | 1.250 | (1.171–1.334) | 0.811 | <0.001 |
| Cys-C | 6 | 0.38 | 1.663 | (1.508–1.834) | 0.735 | <0.001 | 0.983 | (0.869–1.112) | 0.808 | 0.784 |
| SPON1 | 7 | 0.31 | 1.845 | (1.625–2.095) | 0.733 | <0.001 | 1.247 | (1.090–1.427) | 0.811 | 0.001 |
| U-PAR | 8 | 0.35 | 1.565 | (1.394–1.756) | 0.721 | <0.001 | 1.088 | (0.946–1.252) | 0.809 | 0.238 |
| GDF-15 | 9 | 0.79 | 1.726 | (1.547–1.925) | 0.731 | <0.001 | 1.264 | (1.100–1.452) | 0.811 | <0.001 |
| IL-6 | 10 | 0.95 | 1.524 | (1.396–1.664) | 0.726 | <0.001 | 1.133 | (1.021–1.257) | 0.811 | 0.018 |
| HGF | 12 | 0.45 | 1.550 | (1.410–1.704) | 0.721 | <0.001 | 1.263 | (1.137–1.404) | 0.813 | <0.001 |
| OPG | 13 | 0.40 | 1.525 | (1.360–1.710) | 0.715 | <0.001 | 1.290 | (1.139–1.460) | 0.811 | <0.001 |
| TGF-alpha | 14 | 0.44 | 1.288 | (1.201–1.381) | 0.713 | <0.001 | 1.073 | (0.964–1.195) | 0.808 | 0.198 |
| TNF-R1 | 15 | 0.40 | 1.514 | (1.356–1.692) | 0.715 | <0.001 | 1.012 | (0.864–1.185) | 0.808 | 0.884 |
| MMP-12 | 16 | 0.74 | 1.428 | (1.289–1.581) | 0.711 | <0.001 | 1.098 | (0.973–1.238) | 0.810 | 0.129 |
| CHI3L1 | 18 | 1.17 | 1.432 | (1.314–1.561) | 0.716 | <0.001 | 1.180 | (1.065–1.307) | 0.811 | 0.002 |
| LIF-R | 19 | 0.28 | 1.472 | (1.304–1.660) | 0.713 | <0.001 | 1.102 | (0.970–1.252) | 0.808 | 0.137 |
| FGF-23 | 20 | 0.75 | 1.344 | (1.242–1.454) | 0.716 | <0.001 | 1.132 | (1.034–1.239) | 0.810 | 0.007 |
| CCL25 | 22 | 0.64 | 1.399 | (1.244–1.573) | 0.710 | <0.001 | 1.157 | (1.024–1.308) | 0.810 | 0.020 |
| WBC (lab log2) | 23 | 0.38 | 1.336 | (1.211–1.473) | 0.711 | <0.001 | 1.155 | (1.036–1.289) | 0.811 | 0.009 |
| CCL28 | 24 | 0.44 | 1.194 | (1.103–1.293) | 0.701 | <0.001 | 1.077 | (0.966–1.202) | 0.809 | 0.181 |
| LOX-1 | 25 | 0.76 | 1.184 | (1.090–1.285) | 0.702 | <0.001 | 1.113 | (1.015–1.220) | 0.810 | 0.022 |
| Lp-PLA2 (lab log2) | 27 | 0.43 | 1.287 | (1.160–1.428) | 0.699 | <0.001 | 1.159 | (1.039–1.292) | 0.810 | 0.008 |
| TIM | 28 | 0.91 | 1.395 | (1.272–1.530) | 0.713 | <0.001 | 1.162 | (1.049–1.287) | 0.810 | 0.004 |
| sST2 | 29 | 0.59 | 1.509 | (1.354–1.682) | 0.714 | <0.001 | 1.199 | (1.079–1.332) | 0.812 | <0.001 |
| CA-125 | 31 | 0.92 | 1.466 | (1.318–1.629) | 0.713 | <0.001 | 1.155 | (1.047–1.275) | 0.809 | 0.004 |
| REN | 36 | 0.99 | 1.237 | (1.110–1.378) | 0.701 | <0.001 | 1.209 | (1.089–1.343) | 0.811 | <0.001 |
| GH | 37 | 2.00 | 1.322 | (1.201–1.455) | 0.703 | <0.001 | 1.101 | (0.989–1.225) | 0.808 | 0.079 |

Values are NPX values (log2 scale) for the PEA biomarkers and log2 for the levels of NTproBNP, TroponinT-hs, IL-6, GDF-15, Cys-C, CRP-hs, Lp-PLA2 activity, and WBC measured by conventional quantitative assays. SD, standard deviation. HRs and 95% CI are calculated for increase of 1 SD. HR and C-indices are calculated after adjustment for clinical variables as above and also after adjustment for the biomarkers NTproBNP, cTnT-hs, and Cys-C.

a Clinical variable adjustment includes the variables age, sex, BMI, smoking, hypertension, diabetes mellitus, prior MI, prior PCI or CABG, prior stroke/TIA, prior PAD, and randomized treatment.
b Adjustment also for Cys-C, NTproBNP, and cTnT-hs affects the HRs and C-indices of all variables. This leads to all C-indices having a minimal level of 0.808, i.e., the C-index for clinical variables and the 3 biomarkers Cys-C, NTproBNP, and cTnT-hs. The incremental discriminatory value of adding any additional biomarker can then be estimated by subtracting the corresponding C-index with 0.808, e.g., for GDF-15 the increment of the C-index is 0.003.

BMI, body mass index; BNP, brain natriuretic peptide; CA-125, carbohydrate antigen 125; CABG, coronary artery bypass grafting; CCL25, chemokine ligand 25; CCL28, chemokine ligand 28; CHI3L1, chitinase-3 like protein 1; CI, confidence interval; cTnT-hs, high-sensitivity troponin T; CV, cardiovascular; Cys-C, cystatin-C; FGF23, fibroblast growth factor 23; GDF-15, growth differentiation factor 15; HGF, hepatocyte growth factor; GH, growth hormone; HR, hazard ratio; IL-6, interleukin 6; LIF-R, leukemia inhibitory factor receptor; LOX1, lectin-like oxidized LDL receptor 1; Lp-PLA2, lipoprotein associated phospholipase A2; MI, myocardial infarction; MMP-12, matrix metalloproteinase-12; NPX, normalized protein expression; NTproBNP, N-terminal prohormone of brain natriuretic peptide; OPG, osteoprotegerin; PAD, peripheral artery disease; PCI, percutaneous coronary intervention; PEA, proximity extension assay; REN, renin; RF, Random Survival Forest; SD, standard deviation; SPON1, Spondin 1; sST2, soluble suppression of tumorigenesis 2 protein; STABILITY, STabilization of Atherosclerotic plaque By Initiation of darapLadIb TherapY; TGF-alpha, transforming growth factor alpha; TIA, transient ischemic attack; TIM, transmembrane immunoglobulin and mucin domain protein; TNF-R1, tumor necrosis factor receptor 1; TRAIL-R2, tumor necrosis factor-related apoptosis-inducing ligand receptor 2; U-PAR, urokinase plasminogen activator receptor; VEGF-D, vascular endothelial growth factor D; WBC, white blood cell.

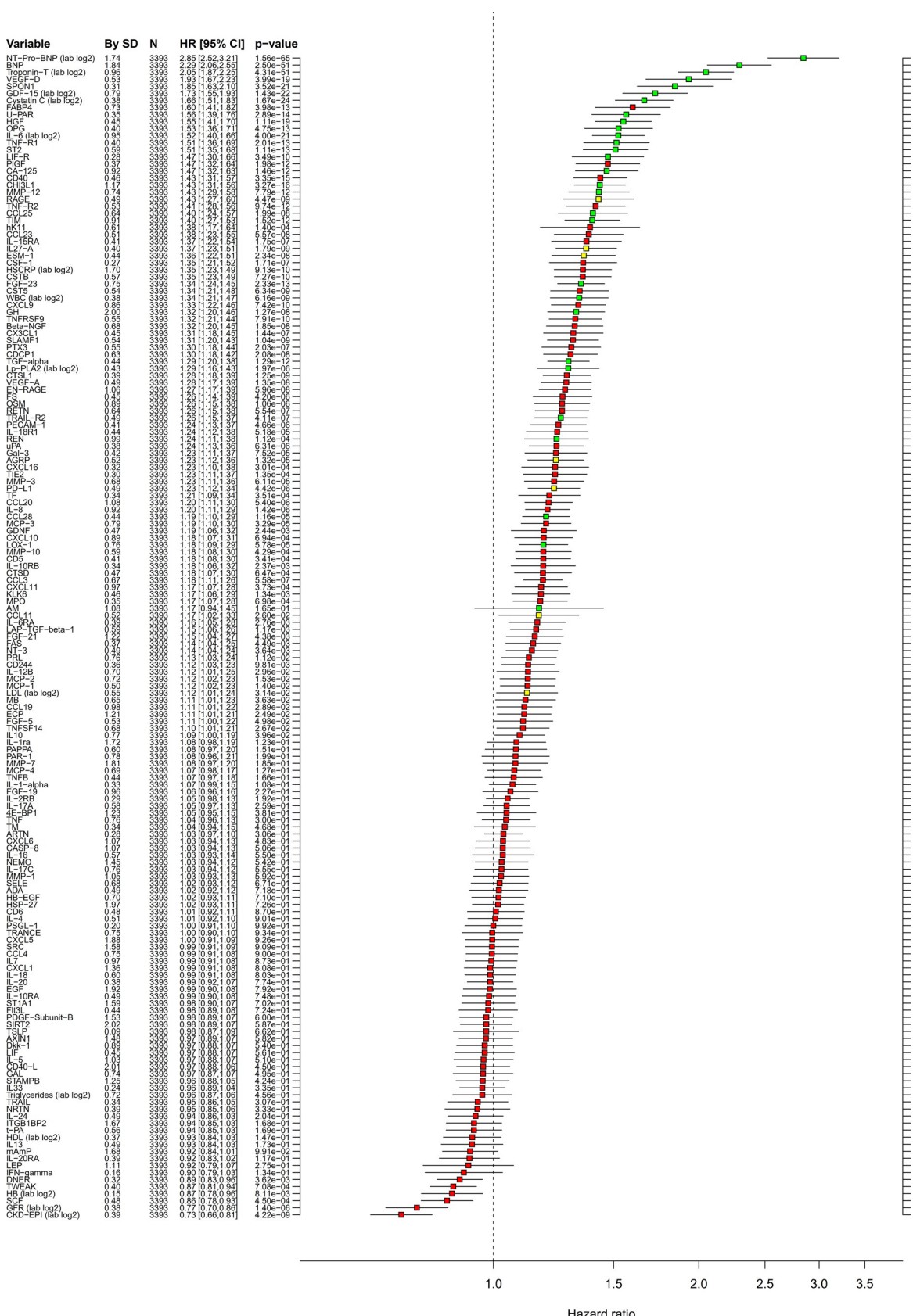

**Fig 2. Multivariable adjusted linear associations in the STABILITY cohort.** Cox regression analyses in the STABILITY cohort of associations between biomarkers and CV death with adjustment for baseline characteristics. Values are the same as for Fig 1. HRs and 95% CI are calculated for increase of 1 SD. Color coding according to the Boruta analysis result: green = confirmed, yellow = tentative, and red = rejected. CI, confidence interval; CV, cardiovascular; HR, hazard ratio; SD, standard deviation; STABILITY, STabilization of Atherosclerotic plaque By Initiation of darapLadIb TherapY.

## Discussion

The present study investigated the associations between multiple circulating protein biomarkers, measured by immunoassays and PEA, and CV death in 2 large cohorts of optimally treated patients with chronic CHD. Employing machine learning approach, 18 biomarkers had confirmed and validated associations with CV mortality. In both cohorts, the strongest associations were found with NTproBNP and cTnT-hs, and also, the previously established independent associations with GDF-15, IL-6, and Cys-C were verified. The novel findings in the study was the identification of 13 additional proteins with an independent association to CV mortality in patients with chronic CHD, i.e., TIM-1, REN, OPG, sST2, TRAIL-R2, CA-125, BNP, HGF, SPON-1, FGF-23, CHI3L1, TNF-R1, and AM. The results indicate that myocardial strain–dysfunction–hypertrophy–fibrosis (NTproBNP, BNP, sST-2, SPON-1, and CA-125), myocyte death and apoptosis (cTnT-hs, TNF-R1, and TRAIL-R2), kidney injury (Cys-C, FGF-23, and TIM-1), hemodynamic stress, renin–angiotensin system (RAS) activation (REN and AM), oxidative stress and inflammation (GDF-15, IL-6, OPG, and CHI3L1), and angiogenesis and vascular cell proliferation (HGF) are important mechanisms associated with CV death in patients with chronic CHD (Fig 5).

The current study confirmed previous findings from our group and others on the very strong importance of the cardiac biomarkers NTproBNP and cTnT-hs for prediction of CV

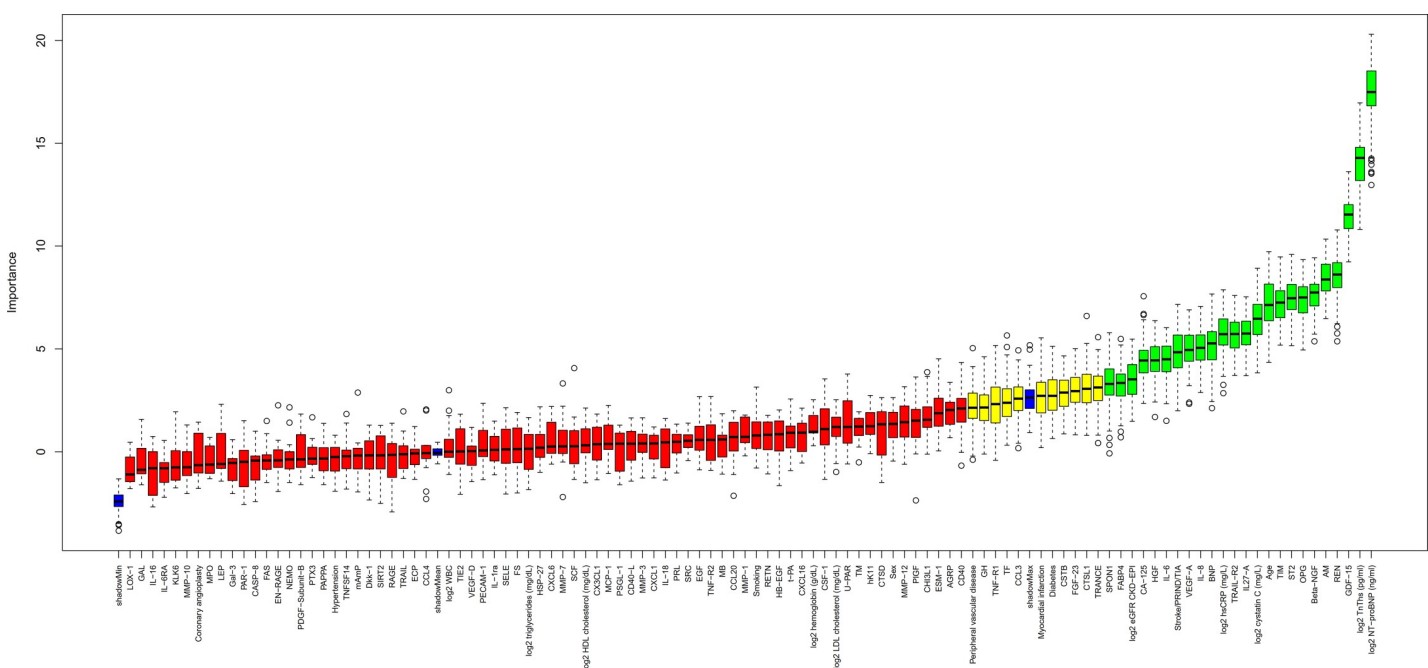

**Fig 3. Variable importance in the replication cohort.** Boruta analysis in the LURIC cohort of the significance of variable importance for CV death in the RF analysis, including clinical variables as well as established and PEA biomarkers. Values are the same as for Fig 1. Color coding according to the Boruta analysis result: green = confirmed, yellow = tentative, and red = rejected. CV, cardiovascular; LURIC, Ludwigshafen Risk and Cardiovascular Health; PEA, proximity extension assay; RF, Random Survival Forest.

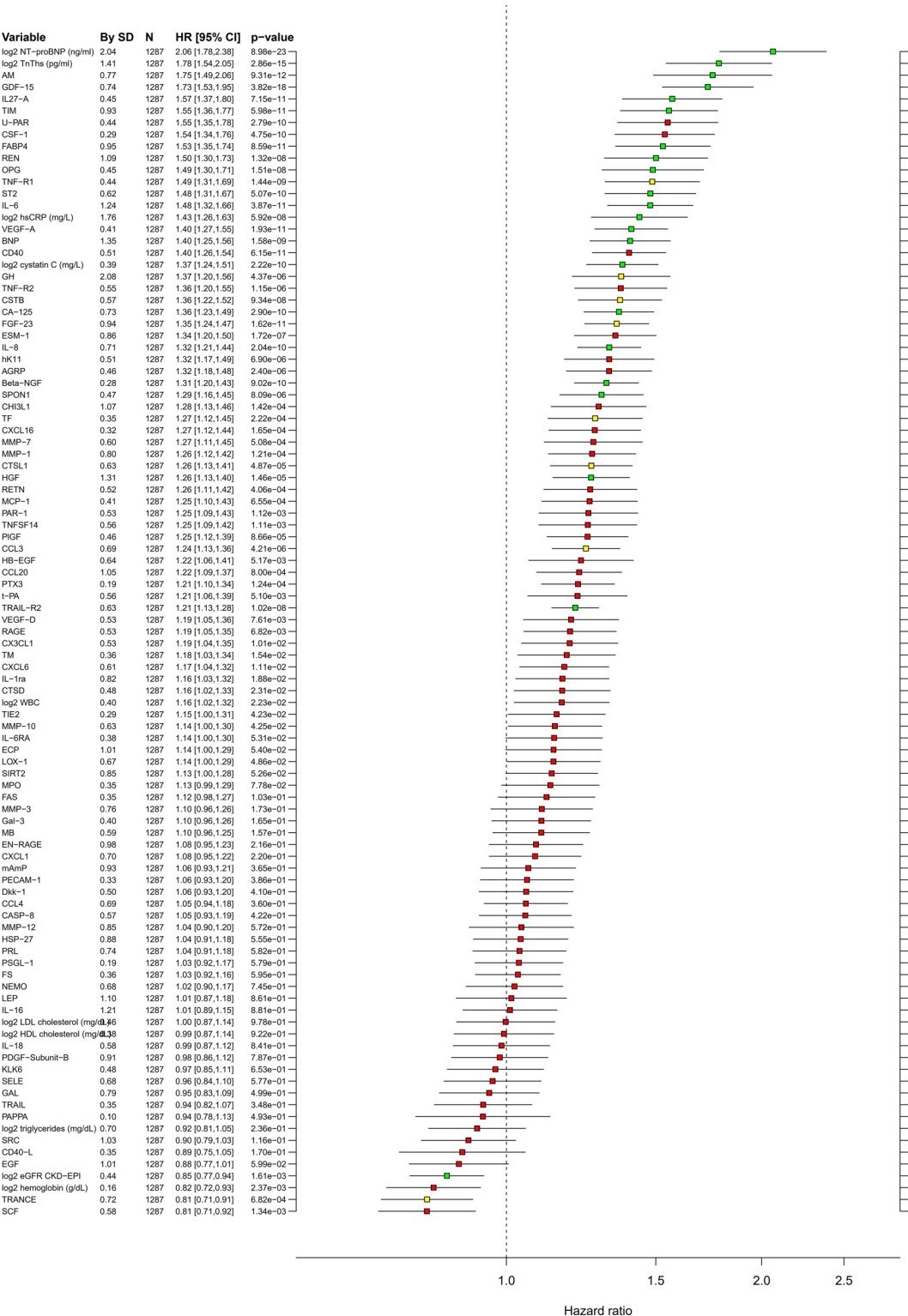

**Fig 4. Multivariable adjusted linear associations in the replication cohort.** Cox regression analyses in the LURIC cohort of associations between biomarkers and CV death after adjustment for baseline characteristics. Values are the same as for Fig 1. HRs and 95% CI are calculated for increase of 1 SD. Color coding according to the Boruta analysis result: green = confirmed, yellow = tentative, and red = rejected. CI, confidence interval; CV, cardiovascular; HR, hazard ratio; LURIC, Ludwigshafen Risk and Cardiovascular Health; SD, standard deviation.

**Table 3. Associations between CV death and biomarkers including biomarkers with confirmed importance according to the Boruta RF analyses in both cohorts.**

| Biomarker | Order RF | Adjusted by clinical variables[a] | | | | | Also for Cys-C+NTproBNP+cTnT-hs[b] | | | |
|---|---|---|---|---|---|---|---|---|---|---|
| | | By SD | HR | CI | C-index | p-value | HR | CI | C-index | p-value |
| NTproBNP | 1 | 2.04 | 2.079 | (1.799–2.402) | 0.785 | <0.001 | 1.779 | (1.495–2.117) | 0.793 | <0.001 |
| cTnT-hs | 2 | 1.42 | 1.715 | (1.491–1.973) | 0.761 | <0.001 | 1.266 | (1.065–1.505) | 0.793 | 0.008 |
| GDF-15 | 3 | 0.74 | 1.728 | (1.527–1.955) | 0.765 | <0.001 | 1.397 | (1.180–1.654) | 0.800 | <0.001 |
| AM | 4 | 0.77 | 1.750 | (1.490–2.056) | 0.761 | <0.001 | 1.352 | (1.126–1.623) | 0.797 | 0.001 |
| OPG | 5 | 0.45 | 1.488 | (1.297–1.708) | 0.747 | <0.001 | 1.256 | (1.087–1.452) | 0.799 | 0.002 |
| TIM-1 | 6 | 0.93 | 1.555 | (1.362–1.775) | 0.748 | <0.001 | 1.294 | (1.123–1.491) | 0.797 | <0.001 |
| REN | 7 | 1.09 | 1.501 | (1.305–1.727) | 0.747 | <0.001 | 1.523 | (1.329–1.745) | 0.806 | <0.001 |
| Cys-C | 8 | 0.39 | 1.370 | (1.243–1.510) | 0.747 | <0.001 | 1.066 | (0.951–1.194) | 0.793 | 0.274 |
| TRAIL-R2 | 9 | 0.63 | 1.205 | (1.131–1.285) | 0.740 | <0.001 | 1.100 | (0.990–1.223) | 0.794 | 0.076 |
| sST2 | 10 | 0.62 | 1.478 | (1.307–1.672) | 0.745 | <0.001 | 1.250 | (1.100–1.422) | 0.796 | <0.001 |
| BNP | 14 | 1.35 | 1.399 | (1.255–1.561) | 0.756 | <0.001 | 0.860 | (0.738–1.001) | 0.792 | 0.052 |
| HGF | 17 | 1.31 | 1.259 | (1.134–1.396) | 0.737 | <0.001 | 1.157 | (1.031–1.299) | 0.795 | 0.013 |
| IL-6 | 18 | 1.24 | 1.478 | (1.316–1.659) | 0.752 | <0.001 | 1.261 | (1.110–1.432) | 0.798 | <0.001 |
| CA-125 | 20 | 0.69 | 1.347 | (1.226–1.479) | 0.748 | <0.001 | 1.101 | (0.988–1.228) | 0.795 | 0.082 |
| SPON1 | 21 | 0.47 | 1.295 | (1.156–1.450) | 0.741 | <0.001 | 1.072 | (0.932–1.232) | 0.794 | 0.331 |
| TNF-R1 | 22 | 0.44 | 1.486 | (1.307–1.689) | 0.747 | <0.001 | 1.195 | (0.969–1.473) | 0.794 | 0.096 |
| FGF23 | 26 | 0.94 | 1.349 | (1.237–1.472) | 0.749 | <0.001 | 1.109 | (0.978–1.257) | 0.795 | 0.107 |
| CHI3L1 | 34 | 1.07 | 1.284 | (1.129–1.461) | 0.732 | <0.001 | 1.110 | (0.961–1.281) | 0.794 | 0.157 |

The biomarkers are ordered by the position in the RF analysis in LURIC. The Cox regression and C-index results are from the replication cohort from LURIC (*n* = 1,287).

Values are NPX values (log2 scale) for the PEA biomarkers and log2 for the levels of NTproBNP, Troponin-T, Cys-C, and CRP-hs measured by conventional quantitative assays. SD, standard deviation. HR and 95% CI are calculated for increase of 1 SD. C-indices are calculated after adjustment for clinical variables as above and also after adjustment for the biomarkers NTproBNP, troponin-T, and Cys-C.

a Clinical variable adjustment includes the variables age, sex, BMI, smoking, hypertension, diabetes mellitus, prior MI, prior PCI or CABG, prior stroke/TIA, prior PAD, and randomized treatment.

b Adjustment also for Cys-C, NTproBNP, and cTnT-hs affects the HRs and C-indices of all variables. This leads to all C-indices having a minimal level of 0.793, i.e., the C-index for clinical variables and the 3 biomarkers Cys-C, NTproBNP, and cTnT-hs. The incremental discriminatory value of adding any additional biomarker can then be estimated by subtracting the corresponding C-index with 0.793, e.g., for GDF-15 the increment of the C-index is 0.007.

AM, adrenomedullin; BMI, body mass index; CA-125, carbohydrate antigen 125; CABG, coronary artery bypass grafting; CHI3L1, chitinase-3 like protein 1; CI, confidence interval; cTnT-hs, high-sensitivity troponin T; CV, cardiovascular; Cys-C, cystatin-C; FGF23, fibroblast growth factor 23; GDF-15, growth differentiation factor 15; HGF, hepatocyte growth factor; HR, hazard ratio; IL-6, interleukin 6; LURIC, Ludwigshafen Risk and Cardiovascular Health; MI, myocardial infarction; NPX, normalized protein expression; NTproBNP, N-terminal prohormone of brain natriuretic peptide; OPG, osteoprotegerin; PAD, peripheral artery disease; PCI, percutaneous coronary intervention; PEA, proximity extension assay; REN, renin; RF, Random Survival Forest; SD, standard deviation; SPON1, Spondin 1; sST2, soluble suppression of tumorigenesis 2 protein; TIA, transient ischemic attack; TIM-1, transmembrane immunoglobulin and mucin domain protein; TNF-R1, tumor necrosis factor receptor 1; TRAIL-R2, tumor necrosis factor-related apoptosis-inducing ligand receptor 2.

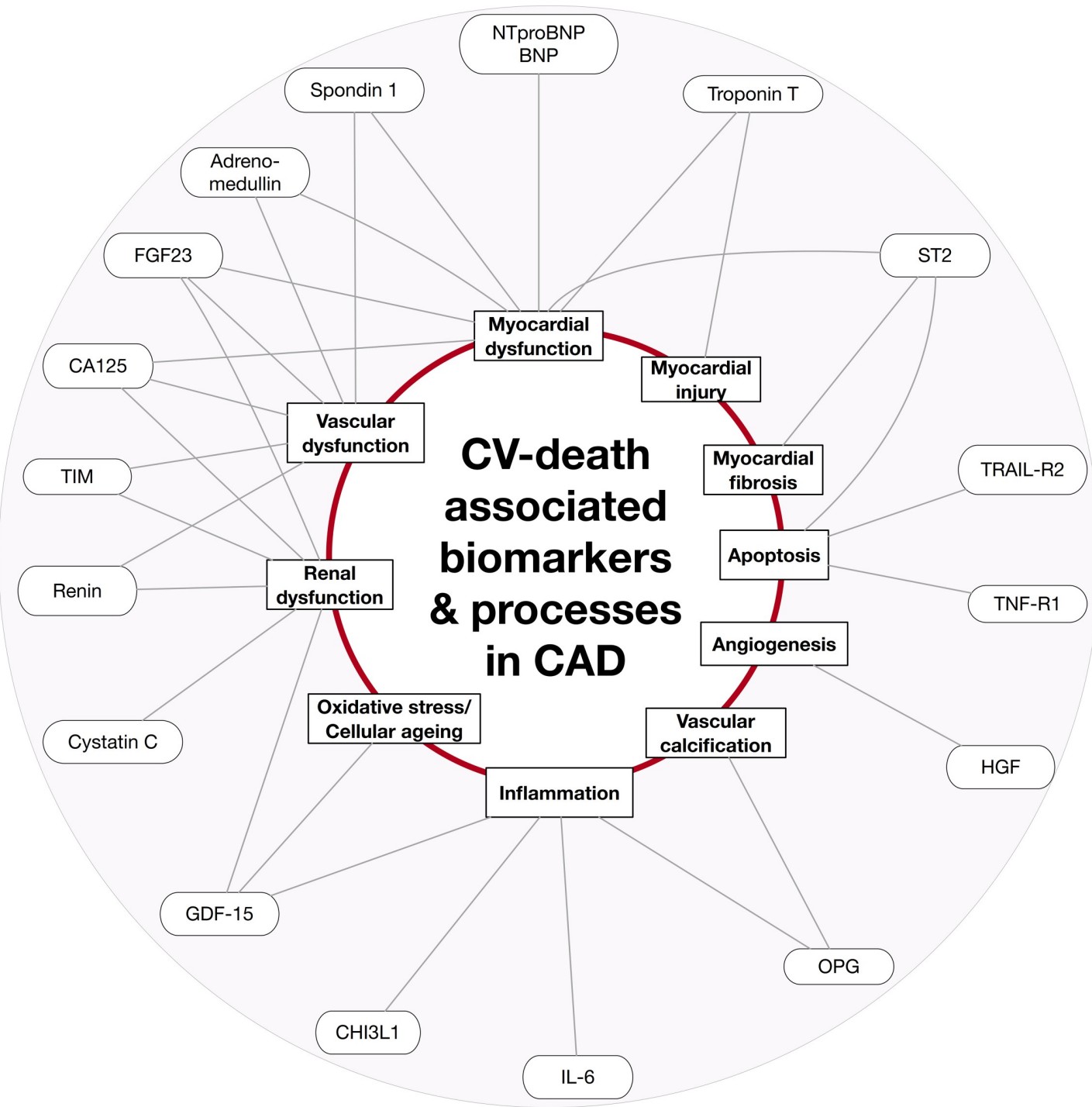

**Fig 5. Biomarkers and processes associated with CV death.** Conceptual figure of biomarkers and processes associated with CV death in chronic CAD. CAD, coronary artery disease; CV, cardiovascular.

death in patients with CHD [6,16,21]. In the present study, these biomarkers were shown to be prognostically more important than any clinical characteristic and more important than 150 other CV and inflammatory biomarkers. NTproBNP and/or BNP and cTnT-hs are related to underlying functional myocardial disturbances, such as myocardial stretch, overload,

dysfunction, hypertrophy, and necrosis. However, no association has been identified between the levels of NTproBNP and cTnT-hs and disease development. Therefore, although associations between other biomarkers and CV death are attenuated by statistical adjustment for NTproBNP and/or cTnT-hs levels and renal dysfunction, still these associations might reflect important pathophysiological pathways.

Multiple biomarker profilings using PEA protein panels have previously been found useful for predicting chronological age using concentrations of 77 plasma proteins in 976 healthy individuals [22,23]. When the OLINK CVD I panel was used in 931 community-dwelling subjects, 7 proteins (OPG, TIM-1, GDF-15, matrix metalloproteinase-12 [MMP-12], REN, TNF ligand superfamily member 14 [TNFSF14], and growth hormone [GH]) were significantly related to the number of carotid arteries affected by plaques after adjustment for multiple testing [24]. From the same study, it was recently reported that 9 proteins (GDF-15, TIM-1, TRAIL-R2, SPON1, MMP-12, follistatin [FS], soluble urokinase-type plasminogen activator surface receptor [sU-PAR], OPG, and sST2) were associated with the development of heart failure [25]. It is noteworthy that 4 of these 9 proteins (GDF-15, TIM-1, OPG, and sST2) were identified and validated in the present study and that 4 others (TRAIL-R2, SPON1, MMP-12, and sU-PAR) were significant in the STABILITY cohort of the present study. The OLINK CVD I panel was also used in 847 patients with acute MI in whom the levels of GDF-15 and TRAIL-R2 were independently associated with all-cause mortality during 7 years follow-up in accordance with the present findings in the STABILITY study [26].

In 2016, a multimarker tool based on estimation of protein levels by the Somalogic modified aptamer technology was used to identify the 4-year risk of MI, stroke, heart failure, and all-cause death in 1 derivation and 1 replication cohort of patients with chronic CAD [27]. This study identified a 9-protein model providing a c-statistic of 0.71 in the replication sample, in which troponin was the only biomarker in common with the present findings. Based on the Luminex xMAP platform, another multimarker model to identify the risk of CV death, MI, or stroke was developed in a derivation cohort of 649 and replication cohort 278 patients. This model included the 4 biomarkers NTproBNP, TIM-1, osteopontin, and tissue inhibitor of metalloproteinase-1 (TIMP-1), which had a c-index of 0.79. Interestingly, NTproBNP and TIM-1 are the same markers as identified in the present study and osteopontin protein may reflect the same processes as OPG [28]. The lack of complete replication of identified markers between the studies might relate to many factors, e.g., differences in measured biomarkers, handling and age of biosamples, assay technologies, endpoints, adjustments, patient populations, treatments, and follow-up.

After the specific cardiac markers, GDF-15, a marker of oxidative stress, cellular aging, and inflammatory activity, showed the strongest association with CV death. The independent prognostic value GDF-15 [5,15,29] concerning total and CV mortality has been documented previously from the STABILITY trial as well as in several other studies. Several other proteins reflecting inflammatory activity (IL-6, CHI3L1, TIM-1, and OPG) were also significantly associated with the risk of CV death in both the present cohorts. Independent associations between IL-6 [17] and CV mortality have previously been shown in this and other studies. Also, CHI3L1 [30] and TIM-1 [31] have been found associated with the development of coronary atherosclerosis. OPG, a member of the tumor necrosis factor receptor superfamily, has pleiotropic effects on bone metabolism, endocrine functions, and vascular inflammation [32,33]. It is expressed by inflammatory stimuli, during acute MI and heart failure, and OPG plasma levels are related to outcomes in patients with these conditions [34–36]. The recently reported simultaneous reduction of inflammatory activity, e.g., as documented by simultaneous reduction of IL-6 levels and ischemic events by treatment with the anti-inflammatory agents canakinumab [37] and colchicine [38], further supports inflammatory activity as a major pathophysiological mechanism in CHD [39].

Chronic kidney disease is a well-established risk factor for CV death. Accordingly, the level of Cys-C, which reflects glomerular filtration rate, has repeatedly been demonstrated to be associated with CV events and death in patients with CHD. The renal hormone renin is a pivotal component of the RAS, which plays a key role in the maintenance of blood pressure and electrolyte-volume homeostasis. The RAS is activated in hypertension and heart failure, and reduction of its activity has become the basis for both prevention and treatment of these conditions [1,2]. AM [40] and CA-125 [41] are indicators of fluid overload and congestion and probably thereby indicators of outcome in CAD. FGF23, a hormone for regulation of phosphate hemostasis and the renin–angiotensin–aldosterone system, is an indicator of both renal and myocardial dysfunction and is associated with left ventricular hypertrophy and clinical outcomes [42]. An animal study has demonstrated *Spon1* mRNA expression in different vascular tissues with the same expression in kidney and heart [43]. In this rat model, the *Spon1* gene was also identified as a novel candidate gene for hypertension. In previous human studies, SPON1 has turned out to be an indicator of myocardial and renal dysfunction [25]. We also confirmed an independent association between TIM-1, a modulator of the T cell–mediated immune response, and CV outcomes [44–46]. Moreover, recently, an anti-TIM-1 antibody was shown to attenuate atherosclerosis development [31]. The identification of all these markers of CV–renal dysfunction as independent risk indicators in the current cohorts of chronic CHD is impressive not least after adjustment for the established and strong biomarkers of renal (Cys-C) and cardiac function (NTproBNP and cTnT-hs) [47].

In the present study, sST2 emerged as an independent biomarker contributing to risk stratification for CV death. ST2 is the receptor for interleukin-33, a cytokine with antihypertrophic and antifibrotic effects on the myocardium. Serum levels of the soluble form of sST2 is a biomarker for myocardial strain and is well established to provide prognostic information in patients with heart failure [48] but also in patients with chronic CHD [49], acute coronary syndrome [50], and in population-based cohorts (Dallas Heart) [51]. The prognostic importance of sST2 in the LURIC cohort has been shown earlier by using a conventional immunoassay, which strengthens the current findings with PEA technology [52].

HGF also appeared as a factor that significantly contributed to the identification of the risk of CV death in the current study. HGF has pleitropic cell functions including angiogenesis, anti-apoptosis, proliferation, and differentiation. Based on high levels in the early phase of MI and in heart failure, HGF has been suggested as a prognostic and diagnostic biomarker of CV disease [53]. HGF gene therapy has also been used in the treatment of ischemic heart disease and for tissue regeneration [54]. In accordance with our findings, the level of HGF has previously been found to be independently associated with the progression of atherosclerosis and clinical events in patients with CHD and heart failure and with long-term mortality in the general population [55–59].

Finally, TNF-R1 and TRAIL-R2 are members of the TNF receptor superfamily and involved in the processes of apoptosis. Probably, these processes are activated by several of the other mechanisms associated with a raised risk for CV death and their incremental importance most likely limited [25,60].

## Limitations and strengths

Although being a large trial with global recruitment, the STABILITY cohort might not be fully representative for all patients with chronic CHD. For example, less than 20% of patients were women, smokers, or had prior multivessel disease. Also, the LURIC observational cohort has limitations, for example, because of recruitment in association with a coronary angiogram and a lack of systematic follow-up concerning secondary preventive treatment and nonfatal events.

Therefore, some biomarkers appearing to have prognostic importance in one setting might not be possible to verify because of the differences in design and size of the studies. However, the differences in design might also be considered an advantage when searching for more robust prognostic biomarkers with relevance in the broad real-life patient population. In the LURIC trial, the associations using 4 years or 12 years follow-up were similar, although with weaker significances at 4 years because of fewer events. Evaluation of shorter time follow-up was avoided because of low event rates and thereby too low statistical power for reliable multivariate evaluation of multiple biomarkers. Some biomarkers had low availability, making them impossible to evaluate in this setting.

## Implication and next steps

The employed multiplex PEA methodology provides a profile of the relative protein concentrations, which is appropriate for screening of the most useful proteins to include in a multiplex assay. However, for final evaluation of the usefulness of protein profiling, a multiplex assay allowing simultaneous and precise quantification across the complete dynamic range of all included proteins will be needed.

## Conclusions

In patients with chronic CHD, 18 out of 157 circulating biomarkers had internally confirmed and externally validated independent significant associations with CV death. The results indicate that myocardial strain–dysfunction–hypertrophy–fibrosis (NTproBNP, BNP, sST2, SPON1, and CA-125), myocyte death and apoptosis (cTnT-hs, TNF-R1, and Trail-R2), kidney injury (Cys-C, FGF-23, and TIM-1), hemodynamic stress, RAS activation (REN and AM), oxidative stress (GDF-15 and OPG), vascular inflammation and immune modulation (IL-6 and CHI3L1), and angiogenesis and vascular cell proliferation (HGF) are important mechanisms associated with CV death in patients with chronic CHD (Fig 5). Profiles of levels of multiple plasma proteins can be useful for the identification of different pathophysiologic pathways associated with an increased risk of CV death in patients with chronic CHD. Measurements of these profiles and their pathways might be useful for the identification of new treatment targets and balancing different treatments in patients with chronic CHD.

## Supporting information

**S1 STROBE Checklist. STROBE, Strengthening the Reporting of Observational Studies in Epidemiology.**
(DOCX)

**S1 Text. Statistical analysis plan.**
(PDF)

**S1 Table. Proteins included on the Olink CVD I and Inflammation panels.**
(PDF)

**S2 Table. PEA biomarker NPX values.** (A) Data availability in all 4,127 patients in the STABILITY cohort. (B) Data availability in all 1,331 patients in the LURIC cohort. LURIC, Ludwigshafen Risk and Cardiovascular Health; NPX, normalized protein expression; PEA, proximity extension assay; STABILITY, STabilization of Atherosclerotic plaque By Initiation of darapLadIb TherapY.
(PDF)

**S3 Table. Baseline characteristics in the randomly selected subcohort and in all patients in the STABILITY trial.** STABILITY, STabilization of Atherosclerotic plaque By Initiation of darapLadIb TherapY.
(PDF)

**S4 Table.** (A) Reproducibility of biomarkers determined in both the CVD1 and Inflammation OLINK PEA panels. (B) Comparisons between biomarkers associations with CV death when using measurements with conventional (lab log2) and PEA assays (NPX) as evaluated by univariate Cox regression analyses. CV, cardiovascular; NPX, normalized protein expression; PEA, proximity extension assay.
(PDF)

**S5 Table. Spearman correlation between the NPX values of relevant PEA biomarkers (rows with any correlation >0.29) and the logarithmic transformation of the ng/L levels of established biomarkers (including only the random cohort in the STABILITY cohort).** NPX, normalized protein expression; PEA, proximity extension assay; STABILITY, STabilization of Atherosclerotic plaque By Initiation of darapLadIb TherapY.
(PDF)

**S6 Table. PEA biomarker NPX values in patients with and without CV death in (A) the STABILITY cohort and (B) the LURIC cohort (numbers are percentiles 25th/50th/75th).** CV, cardiovascular; LURIC, Ludwigshafen Risk and Cardiovascular Health; NPX, normalized protein expression; PEA, proximity extension assay; STABILITY, STabilization of Atherosclerotic plaque By Initiation of darapLadIb TherapY.
(PDF)

**S1 Fig. Random Forest analysis in (A) the STABILITY cohort and (B) the LURIC cohort of prognostic variables for CV death including clinical variables as well as established and PEA biomarkers.** Each estimate of variable importance is colored by the Boruta analysis result. CV, cardiovascular; LURIC, Ludwigshafen Risk and Cardiovascular Health; PEA, proximity extension assay; RF, Random Survival Forest; STABILITY, STabilization of Atherosclerotic plaque By Initiation of darapLadIb TherapY.
(PDF)

**S2 Fig. Multivariable unadjusted linear associations by Cox regression analyses in (A) the LURIC cohort and (B) the STABILITY cohort of associations between biomarkers and CV death.** Values are the same as for Fig 1. HR and 95% CI are calculated for increase of 1 SD. Color coding according to the Boruta analysis result: green = confirmed, yellow = tentative, and red = rejected. Variables colored white were not included in the Boruta analysis. CI, confidence interval; CV, cardiovascular; HR, hazard ratio; LURIC, Ludwigshafen Risk and Cardiovascular Health; PEA, proximity extension assay; SD, standard deviation; STABILITY, STabilization of Atherosclerotic plaque By Initiation of darapLadIb TherapY.
(PDF)

## Author Contributions

**Conceptualization:** Lars Wallentin, Niclas Eriksson, Agneta Siegbahn.

**Data curation:** Mikael Åberg.

**Formal analysis:** Lars Wallentin, Niclas Eriksson, Agneta Siegbahn.

**Funding acquisition:** Lars Wallentin.

**Investigation:** Maciej Olszowka, Tanja B. Grammer, Emil Hagström, Claes Held, Marcus E. Kleber, Wolfgang Koenig, Winfried März, Ralph A. H. Stewart, Harvey D. White, Mikael Åberg.

**Project administration:** Lars Wallentin.

**Supervision:** Lars Wallentin.

**Writing – original draft:** Lars Wallentin.

**Writing – review & editing:** Niclas Eriksson, Maciej Olszowka, Tanja B. Grammer, Emil Hagström, Claes Held, Marcus E. Kleber, Wolfgang Koenig, Winfried März, Ralph A. H. Stewart, Harvey D. White, Mikael Åberg, Agneta Siegbahn.

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
