## [Editor Report · Decision Letter 0]

3 Sep 2020

Dear Dr Wallentin, 

Thank you for submitting your manuscript entitled "Profiling of plasma proteins associated with cardiovascular death in patients with chronic coronary heart disease" for consideration by PLOS Medicine.

Your manuscript has now been evaluated by the PLOS Medicine editorial staff [as well as by an academic editor with relevant expertise] and I am writing to let you know that we would like to send your submission out for external peer review.

Kind regards,

Adya Misra, PhD,

Senior Editor

PLOS Medicine

---

## [Decision Letter · Decision Letter 1]

1 Oct 2020

Dear Dr. Wallentin,

Thank you very much for submitting your manuscript "Profiling of plasma proteins associated with cardiovascular death in patients with chronic coronary heart disease" (PMEDICINE-D-20-04160R1) for consideration at PLOS Medicine. 

[LINK]

In light of these reviews, I am afraid that we will not be able to accept the manuscript for publication in the journal in its current form, but we would like to consider a revised version that addresses the reviewers' and editors' comments. Obviously we cannot make any decision about publication until we have seen the revised manuscript and your response, and we plan to seek re-review by one or more of the reviewers. 

We expect to receive your revised manuscript by Oct 22 2020 11:59PM. Please email us (plosmedicine@plos.org) if you have any questions or concerns.

We look forward to receiving your revised manuscript. 

Sincerely,

Adya Misra, PhD

Senior Editor 

PLOS Medicine

plosmedicine.org

Comments from Academic Editor

Tone down the causal nature of the observations. There is too much emphasis on 'mechanisms' and I'd say that these are potential mechanisms. In cardiovascular research, there is little evidence to suggest that risk factors differ for primary or secondary prevention. Can they discuss the implications of restricting the population to those with CHD only? Would one expect a different pattern when investigated in a different at-risk group?

Please revise your title according to PLOS Medicine's style. Your title must be nondeclarative and not a question. It should begin with main concept if possible. "Effect of" should be used only if causality can be inferred, i.e., for an RCT. Please place the study design ("A randomized controlled trial," "A retrospective study," "A modelling study," etc.) in the subtitle (ie, after a colon).

Abstract- please provide brief participant demographics 

Abstract-please provide 95% CI and p-values when describing association

Abstract-The last sentence of the methods and findings section should include 2-3 limitations of your study design/methodology

Abstract conclusions 

Please address the study implications without overreaching what can be concluded from the data; the phrase "In this study, we observed ..." may be useful. * Please interpret the study based on the results presented in the abstract, emphasizing what is new without overstating your conclusions. * Please avoid vague statements such as "these results have major implications for policy/clinical care". Mention only specific implications substantiated by the results. * Please avoid assertions of primacy ("We report for the first time....")

Author summary

Please use square brackets for references, providing punctuation after a single space. For example xxxx [1,2]. 

Methods

Did your study have a prospective protocol or analysis plan? Please state this (either way) early in the Methods section.

Please ensure that the study is reported according to the [STROBE] guideline, and include the completed [STROBE or other] checklist as Supporting Information. When completing the checklist, please use section and paragraph numbers, rather than page numbers. Please add the following statement, or similar, to the Methods: "This study is reported as per the Strengthening the Reporting of Observational Studies in Epidemiology (STROBE) guideline (S1 Checklist)."

Please report your study according to the relevant guideline, which can be found here: http://www.equator-network.org/

Please introduce Lp-PLA2 on first view 

Please provide participant details for both cohorts- including details of where they were recruited, dates of recruitment and follow up. 

Please provide p-values of up to three decimal places throughout the text. Please ensure these are accompanied by 95%CI

Page 17 Line 216 suggest replacing “confirmed importance for CV death” to “associated with CVD death” or similar 

Discussion

Please present and organize the Discussion as follows: a short, clear summary of the article's findings; what the study adds to existing research and where and why the results may differ from previous research; strengths and limitations of the study; implications and next steps for research, clinical practice, and/or public policy; one-paragraph conclusion.

I suggest tempering some of the language in this section, “definitely established as key risk indicators” and “demonstrated to outperform” for example. Since this is an observational study please rephrase to remove causal language

Line 268- please remove the word “causal”

You may wish to rephrase optimal treatment here or explain, that patients were treated with xxx medication. You may wish to include this explicit information in the methods. 

Comments from the reviewers:

Reviewer #1: "Profiling of plasma proteins associated with cardiovascular death in patients with chronic coronary heart disease" aims to characterize significant biomarkers with an association to coronary heart disease (CHD). Two cohorts were involved, a derivation cohort consisting of 605 cases with CV death and 2788 non-death cases from the STABILITY trial, and a validation cohort of 245 cases with CV death and 1042 non-death cases from the LURIC study, with mass measurement of protein biomarkers achieved using Proximity Extension Assay (PEA) technology.

The main methodology was based on statistical analysis via the Boruta method on random survival forests (RF), with RF intended to estimate the prognostic importance of variables, and Boruta to determine their significance. Cox regression analyses were also performed. Known prognostic biomarkers such as NTproBNP & cTnT-hs were confirmed as associated with CHD, together with a number of other proteins, for a total of 18 biomarkers of 157 examined.

These findings are interesting in that there appears to be relatively weak consensus on protein biomarker associations with CHD, from prior work. However, while mostly comprehensively described, some points might be clarified, particularly pertaining to data availability and the methodology used:

1. The STABILITY cohort and LURIC cohorts appear to have widely varying follow-up periods (median 3.7 years, and up to 12 years respectively). It might be commented on whether this warrants additional treatment (e.g. short-term vs. long-term CHD mortality)

2. The Pearson/Spearman correlation coefficients for the PEA biomarker measurements (as referenced in Line 117) might be provided if possible in the supplementary.

3. The relatively low availability of a large number of biomarkers (Table S2a), together with the minimal availability criteria required to be included in further analysis, might be supported in greater detail. For example, the TSLP biomarker is listed as having 99.5% of values below the lower limit of detection, which appears to imply that it qualifies to be considered for assessment of association with CHD, despite having only 1 in 200 subjects having data available for it. While there is no objection against having such biomarker values reported in principle, the qualification for further analyses might be justified (especially as such values appear to be automatically imputed from Line 179, despite the scant basis for such imputation)

4. Further on this, certain biomarkers such as BNP (39.3% availability in STABILITY) appear to be presented as significant (BNP is the third top variable, in Figure S1a), despite relatively low availability. It is not immediately evident whether there might be any bias involved towards the subset of patients with detectable BNP, given that this value was not available for nearly 40% of patients. Moreover, is data availability accounted for in determining significance for the statistical analysis? This might be further commented on.

5. On the C-index, while it is stated that "...the incremental discriminative value of each biomarker was illustrated by the C-index" (Line 175), it is not immediately evident as to how this is arrived at. For example, the C-index for NTproBNP is given as 0.802, in Table 2. Does this C-index reflect the performance of the model given all variables (including/excluding NTproBNP?), and if so, does the C-index of Troponin-T reflect the performance give all variables including/excluding Troponin-T, and possibly also including/excluding NTproBNP? Given the relative novelty of the RF-Boruta analysis, it might be appropriate to devote slightly more text to explaining it (possibly in the supplementary material)

6. The "By SD" column for Tables 2 & 3/Figures 2 & 4 might be initially defined, together with abbreviations such as those for hazard ratio & confidence interval.

7. The biomarker unadjusted values for HR, CI & C-index (as referenced in Line 171) might also be provided as for the adjusted values in Tables 2 & 3.

8. From what could be understood, the derivation STABILITY cohort was first used to develop an RF-Boruta model (and its variable importance analyzed), while the validation LURIC cohort was then used to develop a separate/independent RF-Boruta model, though with input in the form of biomarkers of confirmed importance (Line 216) as determined from the derivation cohort model. The associations for both the derivation and validation cohorts are as presented in Table 3.

However, validation might be more commonly understood as applying a model obtained from the training (derivation) data, to the validation data, and reporting the performance (possibly C-index in this case). This might be considered.

Reviewer #2: The authors Lars Wallentin et al, proposed to reviewer an interesting publication entitled "Circulating biomarkers are associated with development of coronary heart disease (CHD) and its complications by reflecting pathophysiological pathways and/or organ dysfunction".

They explored the associations between cardiovascular (CV) and inflammatory biomarkers and CV death using proximity extension assays (PEA) in 605 cases with CV death and 2,788 randomly selected non-cases during 3 - 5 years follow-up in the STABILITY trial

They seek for validation of their findings in in the LURIC cohort consisted of 245 cases and 1,042 non-cases during 12 years follow-up. 

Authors to conclude that Protein profiles based on clusters of biomarkers reflecting different pathophysiologic mechanisms might be useful for identification of new treatment targets and balancing different treatments in patients with chronic CHD.

While the manuscript is really pleasant to read, there is several points the reviewer which to be seen addressed.

Minor points:

- Trial registration is given for STABILITY study and not for the LURIC one

- The methods previously used to measure NTproBNP, cTnT-hs, CRP-hs, IL-6, GDF-15 and Lp-PLA2 should be mentioned

- Table 1 in the legend "For BMI, weight, and biomarkers…" this is not relevant as BMI, weight are not presented in this table. However, this remark stands for the Age of the patients

- Table S3, presentation by percentils 25th/50th/75th are also relevant for the age

- Table 1, 2, 3 & 4: a color code should be givent to understand the meaning of the red, yellow, blue and green boxes

- Page 23, line 300-303: "The lack of complete replication … relate to many factors e.g…." important to add "handling and age of the biosamples" as the stability overtime of the biosamples could be critical. 

Major points:

- The authors detailed within TableS1 the panel from which the measurements have been used for the 27 redundant biomarkers. It is important to have this description as well in the main tables. Furthermore, it is also important to have a comparison of the results obtained on both panels for those 27 proteins present. What are the reproducibility & variability between the two panels while using the same samples? 

- A comparison of the data obtained with the different methods (conventional & PEA) for the previously used NTproBNP, cTnT-hs, CRP-hs, IL-6, GDF-15 and Lp-PLA2 would be nice to have. The manuscript suggest an interest in using the multiplex method, thus methods should be compared side by side for the above listed biomarkers. 

- Discussion looks very much like other publications using the PEA or other multiplex technologies (Somalogic, …) even if figure 5 gives a nice resume of the findings. The conclusion on Page 27, line 393 suggesting the" identification of potential treatment targets and balancing different treatments, in patients with CHD" is not enough developed. Even if the topic of the journal is Medicine, important information coming from experimental studies at least on some of the listed biomarkers and all listed pathways could enriched the discussion.

Reviewer #3: In this work Wallentin et al. study the association between 157 cardiovascular and inflammatory biomarkers and CV death using proximity extension analysis in patients with stable coronary heart disease.

The derivation cohort consisted of 605 cases (who experienced CV death) and 2788 non-cases (stable CHD patients that survived the 3-5 year follow up).

The validation cohort consisted of 245 cases and 1042 non-cases but the follow up was shorter (1 year). 

Biomarker levels were measured using conventional laboratory immunoassays and/or OLINK panels.

28 biomarkers were prognostic in the derivation cohort (most belonged to the CVD panel and only 4 to the inflammation panel).

After validation, a total of 18 biomarkers were found to associate with CV death in both cohorts, of these the strongest predictors were ntProBNP and cTNT-hs.

This is an interesting and data-rich study which highlights the potential clinical utility of multimarker plasma biomarker panels for personalised risk stratification in CHD, yet I have some concerns. 

#The abstract focuses on chronic CHD, while the STABILITY and LURIC methods section refers to stable CHD; chronic and stable CHD are not exactly the same thing, so stick to one term. 

#How was stable CHD confirmed in the STABILITY/LRIC studies, did all participants have baseline invasive coronary angiography? (describe this in main text to avoid sending readers to the original papers to find out). 

#Define CV death and explain how was it ascertained - death certificates, GP records, national statistics office etc?

#Discuss the potential impact of having differential followup duration in the derivation and validation cohorts - consider running a sensitivity analysis on the derivation cohort to check whether the significant biomarkers persist if the follow up duration is whittled to 1year to match that used in the validation cohort. 

#The author's description of PEA technology in the introduction can be better clarified. It is key to understanding the paper so would be good to explain it at the outset and not later in the discussion. 

"Proximity Extension Assay (PEA) technology, allowing simultaneous measurements of 92 proteins in a 1.0 μl plasma sample by PCR amplification of DNA strands from DNA-labeled antibody pairs"

(from OLINK site: A pair of oligonucleotide-labeled antibodies ("probes") are allowed to pair-wise bind to the target protein present in the sample in a homogeneous assay, with no need for washing. When the two probes are in close proximity, a new PCR target sequence is formed by a proximity-dependent DNA polymerization event...the resulting sequence is subsequently detected and quantified using standard real-time PCR.)

#The section about reproducibility of assays (ln 117) should be moved to the results section, it is a result. 

#Did you consider adjusting for socio-economic position in the regression analyses?

#You randomised for the administration of darapladib in the regressions, but could you explain to the reader what is the expected effect of darapladib Rx in these patients? Are we to expect lower death rates in darapladib-treated patients?

[LINK]

---

## [Decision Letter · Decision Letter 2]

1 Dec 2020

Dear Dr. Wallentin,

Thank you very much for re-submitting your manuscript "Screening of plasma proteins associated with cardiovascular death in patients with chronic coronary heart disease – a retrospective study" (PMEDICINE-D-20-04160R2) for review by PLOS Medicine.

I have discussed the paper with my colleagues and the academic editor and it was also seen again by xxx reviewers. I am pleased to say that provided the remaining editorial and production issues are dealt with we are planning to accept the paper for publication in the journal.

[LINK]

We look forward to receiving the revised manuscript by Dec 08 2020 11:59PM. 

Sincerely,

Adya Misra, PhD

Senior Editor 

PLOS Medicine

plosmedicine.org

Requests from Editors:

Title: please revise to “Plasma proteins associated with cardiovascular death in patients with chronic coronary heart disease: a retrospective study”

Please provide p-values up to three decimal places only

Please use square brackets for references and place the bracket before full stop.

Please add brief participant demographics in the abstract

Please provide full details for ref 19, 31,33 

Discussion- please emphasize the low availability of some markers versus the significant ones to provide context. Along with noting which findings confirm previous work, please also highlight the novel aspects of this study

STROBE checklist- please use paragraph and sections instead of page numbers as these are likely to change

Please add study dates in abstract

 At line 304, "hypothesis-free"? Please revise

At line 429, perhaps "megatrial" can be avoided (e.g., "large trial")

Comments from Reviewers:

Reviewer #1: We thank the authors for addressing most of the concerns from the previous review round. However, the issue of (extremely) low availability of some PEA markers (including significant ones such as BNP) might be explicitly recognized as a limitation/discussed in the text, to help place the findings in context.

Reviewer #2: The reviewer read the revised manuscript and is aknowledging that its concerns were addressed. 

Few minor comments howerever:

-the authors stated to have been using an "agnostic approach by machine learning" (p41/65 line 304). This is partially true, knowing that the proteomic approach was performed on a set of preselected targets known to play a role in cardiovascular diseases and inflammation (Olink PEA-CVD1 & Inflammation panels). Obvisously, theexperimental setting was oriented while the analyses were using machine learning. 

- p42/65 line 324 : "inflammatory biomarkers .. NTproBNP" remove one dot

- the authors have not been relating FGF23 to Renin angiotensin aldosterone system while FGF23 is a known regulator of Renin via its control of the 1-12(OH)2D

Reviewer #3: The revised manuscript is improved and I have no further suggested changes.

[LINK]

---

## [Editor Report · Decision Letter 3]

5 Jan 2021

Dear Dr. Wallentin,

I am writing concerning your manuscript submitted to PLOS Medicine, entitled “Plasma proteins associated with cardiovascular death in patients with chronic coronary heart disease – a retrospective study.”

We have now completed our final technical checks and have approved your submission for publication. You will shortly receive a letter of formal acceptance from the editor.

Kind regards,

PLOS Medicine